# Circulating Monocyte Count as a Surrogate Marker for Ventricular-Arterial Remodeling and Incident Heart Failure with Preserved Ejection Fraction

**DOI:** 10.3390/diagnostics10050287

**Published:** 2020-05-08

**Authors:** Kuang-Te Wang, Yen-Yu Liu, Kuo-Tzu Sung, Chuan-Chuan Liu, Cheng-Huang Su, Ta-Chuan Hung, Chung-Lieh Hung, Chen-Yen Chien, Hung-I Yeh

**Affiliations:** 1Division of Cardiology, Department of Internal Medicine, MacKay Memorial Hospital, Taitung Branch, Taitung 95054, Taiwan; wangkuangte@yahoo.com.tw; 2Critical Care Medicine, Department of Internal Medicine, MacKay Memorial Hospital, Tamsui Branch, Tamsui 25160, Taiwan; yenyu1012@gmail.com; 3Division of Cardiology, Department of Internal Medicine, MacKay Memorial Hospital, Taipei Branch, Taipei 10449, Taiwan; 8905012@gmail.com (K.-T.S.); chsu007@gmail.com (C.-H.S.); hung0787@ms67.hinet.net (T.-C.H.); hiyeh@ms1.mmh.org.tw (H.-I.Y.); 4Department of Medicine, Mackay Medical College, New Taipei City 25245, Taiwan; 5Department of Medical Laboratory Science and Biotechnology, Yuanpei University of Medical Technology, Hsinchu 30015, Taiwan; carrie@ms1.mmh.org.tw; 6Health Evaluation Center, MacKay Memorial Hospital, Taipei 10449, Taiwan; 7Mackay Junior College of Medicine, Nursing and Management, New Taipei City 11260, Taiwan; 8Graduate Institute of Health Care Organization Administration, College of Public Health, National Taiwan University, Taipei 100025, Taiwan; 9Division of Cardiovascular Surgery, Department of Surgery, MacKay Memorial Hospital, Taipei 10449, Taiwan

**Keywords:** leukocyte, monocyte, common carotid artery diameter, heart failure with preserved ejection fraction (HFpEF)

## Abstract

Among 2085 asymptomatic subjects (age: 51.0 ± 10.7 years, 41.3% female) with data available on common carotid artery diameter (CCAD) and circulating total white blood cell (WBC) counts, higher circulating leukocytes positively correlated with higher high sensitivity C-reactive protein (hs-CRP). Higher WBC/segmented cells and monocyte counts were independently associated with greater relative wall thicknesses and larger CCADs, which in general were more pronounced in men and obese subjects (body mass index ≥ 25 kg/m^2^) (all P _interaction_: < 0.05). Using multivariate adjusting models, only the monocyte count independently predicted the left ventricular mass index (LVMi) (ß-Coef: 0.06, *p* = 0.01). Higher circulating WBC, segmented, and monocyte counts and a greater CCAD were all independently associated with a higher risk of heart failure (HF)/all-cause death during a median of 12.1 years of follow-up in fully adjusted models, with individuals manifesting both higher CCADs and monocyte counts incurring the highest risk of HF/death (adjusted hazard ratio: 2.81, 95% CI: 1.57. −5.03, *p* < 0.001; P _interaction_, 0.035; lower CCAD/lower monocyte as reference). We conclude that a higher monocyte count is associated with cardiac remodeling and carotid artery dilation. Both an elevated monocyte count and a larger CCAD may indicate a specific phenotype that confers the highest risk of HF, which likely signifies the role of circulating monocytes in the pathophysiology of heart failure with preserved ejection fraction (HFpEF).

## 1. Introduction

Heart failure (HF) with a relatively preserved systolic function (HFpEF) (ejection fraction (EF) > 50%), traditionally quantified as the ventricular ejection fraction in nearly one half of the clinical heart failure population, mainly affects the elderly population [1,2]. Attenuated myocardial contractility in terms of increased ventricular stiffness, accompanied by structural remodeling in response to a stiffened and remodeled arterial vascular system, is a typical and hallmark feature of HFpEF [3,4,5]. It has been proposed that aging and hypertension-related increases in passive myocardial stiffening, coupled with more central arterial structural remodeling (for example, dilated common carotid artery (CCA)) and an attenuated elastance/functional decline, play a central role in the HFpEF pathophysiology [6,7].

A great body of literature has recently suggested pro-inflammatory signaling and demonstrated myocardial oxidative stress, with both vascular and myocardial involvement, as alternative novel pathological mechanisms of HFpEF [8,9,10]. Although it has remained unproven, metabolic derangement and associated systemic pro-inflammatory signaling have been proposed as the primary causes of the arterial and cardiac structural remodeling that leads to HFpEF [11]. Until recently, the infiltration of activated local inflammatory cells (monocyte and monocyte-activated macrophages) within cardiac tissue has been taken to denote a pathological process mediating cardiac structural remodeling and impaired diastolic function, leading to HF development [12,13]. However, the effect of clinical correlates and the possible prognostic impacts of circulating white blood cell (WBC) count on cardiac and common carotid artery (CCA) remodeling in asymptomatic individuals remain largely unexplored. Therefore, this study aimed to explore these associations in a large population-based cohort.

## 2. Materials and Methods

### 2.1. Study Subjects

Our study population comprised three groups of individuals from an ongoing cardiovascular health survey program, which comprised 2119 asymptomatic study participants from January 2003 to June 2009. Part of the data has been published before [14], with the current study mainly focused on a study period prior to the use of advanced echocardiography imaging (i.e., tissue Doppler imaging). The study design was approved by the local ethical institutional committee (Mackay Memorial Hospital) for a retrospective data analysis, with a waiver of informed consent from the study participants (IRB: 11MMHIS059, 26 May 2011). The data security was guaranteed, and none of the authors had access to patient identifying information before or after the data analysis. In brief, comprehensive physical examinations and routine biochemical laboratory studies were performed. In addition, questionnaires regarding medical histories and clinical symptoms were employed. All the baseline characteristics and detailed anthropometric information including age, height, weight, BMI, waist circumference, and buttock circumference were obtained. Hypertension and diabetes were defined as the presence of any known history or current use of medication associated with the disease. The presence of cardiovascular disease (CVD) was defined as a history of coronary artery disease, cerebrovascular disease, peripheral artery disease, or current usage of associated medications. The standardized sphygmomanometer-defined resting blood pressures were measured and reported under resting status by medical staff blinded to the other test results. The exclusion criteria included atrial fibrillation, current hemodialysis, pacemaker implantation, prior or current HF history, active myocardial ischemia (acute coronary syndrome, myocardial infarction, and refractory angina), anemia, thyroid dysfunction, or an established history of chronic lung diseases or of cardiovascular surgery.

All of the study participants were requested to fast for at least 8 h before venipuncture and blood sampling using a BD Vacutainer SSTTM (Becton Dickinson, Franklin Lakes, NJ, USA) sample collection tube. All the sample collections and analyses were performed in a standardized laboratory located at the center with international accreditation (ISO-15189), in compliance with the Clinical Laboratory Standards Institute guidelines (Specimen Choice, Collection, and Handling; Approved Guideline H18-A3). To ensure diagnostic accuracy, the sample testing was performed in the original tubes to avoid sample mixing within 1 day. All the samples were analyzed in triplicate, and assay measurements were confirmed to be in the linear range using an internal standard. Complete blood counts, including the total white blood cell count (WBC), red blood cell, hemoglobin (Hb), hematocrit (Ht), platelet, and leukocyte subtype percentages (including segmented neutrophil (abbreviated as “segmented”), monocyte, and lymphocyte expressed as %), were determined using an autoanalyzer (Beckman Coulter Counter DXH series, Coulters Corporation, FL, USA). The subtypes of the leukocyte counts were determined as the total white blood cell count × the specific leukocyte subtype percentage. The biochemical laboratory data, including the fasting sugar, were analyzed using a Hitachi 7170 Automatic Analyzer (Hitachi Corporation, Hitachinaka Ibaraki, Japan) with the hexokinase method. The blood creatinine was analyzed by a kinetic colorimetric assay, and the blood cholesterol and triglyceride levels were analyzed by an enzymatic method. The details of the analytical methods are described in our previous publication [15]. The renal function was reported as the estimated glomerular filtration rate (eGFR) using the Modification of Diet in Renal Disease (MDRD) equation.

### 2.2. Determination of Ventricular-Arterial Remodeling

#### 2.2.1. Assessment of Ventricular Remodeling

A two-dimensional echocardiography was performed in the left-sided recumbent, supine position in all subjects using a comprehensive Doppler acquisition (Sonos 5500, Philips, Andover, Massachusetts) equipped with a 2.5–4.0 MHz transducer. The left ventricular (LV) wall thickness (LV septal wall thickness: inter-ventricular septal thickness (IVS); posterior wall thickness: LVPW), end-diastolic diameter (LVIDd), and subsequent LV mass (LVM) information (with and without index to the body surface area as the LV mass index (LVMi)) by linear method, along the with chamber-level left ventricular volume and derived ejection fraction (LVEF) were all determined using standardized methods recommended by the American Society of Echocardiography (ASE) practice guidelines [16]. All the echocardiography measurements were averaged over three consecutive heart cycles. The left ventricular hypertrophy (LVH) was defined as an LV mass index (LVMi) greater than 115 gm/m^2^ for men and 95 gm/m^2^ for women. The patients with relative wall thicknesses (RWT) > 0.42 and LVMis within the normal range were classified as having concentric remodeling. Echocardiography information was available for 1805 study participants.

#### 2.2.2. Assessment of Carotid Arterial Remodeling

The extracranial carotid arteries were assessed with high-resolution ultrasound scanners, either Acuson Aspen (Siemens Medical Systems) or Logiq 7 (GE Medical Systems) equipped with a 5–10 MHz linear-array transducer. The carotid images were obtained by a trained and certified vascular technologist. The ultrasonography evaluations were performed in a room at a comfortable temperature following 5–10 min of rest in the supine position. As in our previous report, using a leading-edge-to-leading-edge technique, all the subjects had an arterial lumen measurement at the far wall of the distal 1 cm of each common carotid artery (CCA), carotid bifurcation, and proximal 1 cm of each internal carotid artery (ICA), with the CCA diameter (CCAD) averaged from both sides used as the representative data for each study participant [6]. In the evaluation of the presence or absence of carotid plaques, the full extra-cranial arterial bed was examined using both transverse and longitudinal views.

### 2.3. Determination of the Serum Biomarker, N-Terminal Pro-Brain Natriuretic Peptide (Nt-ProBNP), and the Inflammatory Marker, High Sensitivity C-Reactive Protein (hs-CRP)

The levels of Nt-ProBNP were blinded to the patients’ details. They were determined by an electrochemiluminescence immunoassay (Roche Diagnostics GmbH, D-68298 Mannheim, Germany), based on competitive ligand binding containing two polyclonal antibodies that recognize epitopes located in the N-terminal part (residues 1–76) of proBNP (residues 1–108) in a sandwich format. The inflammatory marker high sensitivity C-reactive protein (hs-CRP) levels were determined by a highly sensitive, latex particle-enhanced immunoassay (Elecsys 2010; Roche Diagnostics GmbH, Mannheim, Germany).

### 2.4. Statistical Analysis

Continuous data are expressed as mean and standard deviation, with categorical data expressed as the frequency and proportion of the occurrence in all subjects. Differences in the baseline demographics among groups were analyzed using Student’s t-test, with categorical data analyzed by chi-squared or Fisher’s exact test as appropriate (Table 1). Univariate and multivariate linear regression models were constructed to determine the independent associations of various leukocyte counts (including total white blood cell count (WBC), segmented cell count, and monocyte and lymphocyte counts, separately, in different models) with inflammatory marker hs-CRP along with diverse ventricular and carotid arterial remodeling (CCAD) information. The individual coefficient value (as ß-Coef) and level of statistical significance (*p* value) of these results were reported (Table 2). Restricted cubic spline (RCS) curves were constructed to explore the pattern of relationships between various leukocyte counts and CCAD (Figure 1). A subgroup analysis regarding the association of CCAD with various leukocyte counts was performed (Figure 2). The potential prognostic utilization (composite HF hospitalization and all-cause death) of CCAD and various leukocyte count groups were tested along with conventional cardiovascular risks (including age; sex; body mass index; systolic blood pressure; biochemical information of fasting sugar and lipid profiles; and a medical history of hypertension, diabetes, known cardiovascular disease (CVD), or active smoking status) by a backward stepwise regression analysis (Table 3). The risk of HF hospitalization based on CCAD and various leukocyte fractions were further examined with adjustment and presented as odds ratios and 95% confidence intervals (CIs) (Figure 2). Kaplan–Meier curves were generated to illustrate the survival trend between various leukocyte/CCAD categories (by a median value of CCAD as: ≤ 7 vs. > 7 mm as lower vs. higher group; various leukocyte count groups as lower vs. higher by median values, respectively) (Figure 3), and Cox linear regression models with (multivariate) and without (univariate) adjustment were conducted to examine the association of various leukocyte/CCAD categories with outcomes (Table 4).

All data were analyzed using the commercial software STATA 11.0 package (Stata-Corp., College Station, TX, USA). The significance level (α-value) for all analyses was two-sided, and 0.05 indicated statistical significance.

## 3. Results

### 3.1. Baseline Demographics and Cardiac Remodeling with CCAD

Individuals with greater CCADs were of more advanced age, more likely to be male, and had higher blood pressures (either systolic/diastolic, or pulse pressure), as well as a greater body weight, waist circumference, and body mass index (all trends, *p* < 0.001) (Table 1). More unfavorable lipid profiles, higher uric acid levels, and worse renal function in terms of lower eGFR were associated with a greater CCAD (all trends, *p* < 0.05). Patients with a higher CCAD also had significantly higher total WBC, hs-CRP, segmented, and monocyte counts, with the highest Nt-ProBNP level observed in the largest CCAD quartile (all trends, *p* < 0.001). Finally, the subjects with greater CCADs were more likely to have hypertension, diabetes, cardiovascular diseases, and an active smoking habit (all trends, *p* < 0.001), and those that regularly exercised demonstrated higher CCADs (trend *p* = 0.021).

Individuals with a greater CCAD were associated with greater wall thickness (either septal (IVS) or posterior wall (LVPW)), larger LV dimension, greater LV mass and LV mass index (LVMi), higher risk of LVH, were more likely to have concentric remodeling (all trends *p* < 0.001), and had a slightly lower LVEF (trend *p*: 0.038) (Table 1). By using stepwise regression models with various blood pressure components (systolic, diastolic, and pulse pressure in separate models), more advanced age, male sex, a larger BMI, higher various blood pressure components (systolic, diastolic, and pulse pressure), a history of hypertension, gout, active smoking, and higher fasting sugar levels were all significantly associated with a larger CCAD (all *p* < 0.05, Appendix A).

### 3.2. Association of Various Leukocyte Counts with the Inflammatory Marker, CCAD, and Cardiac Remodeling

Overall, higher leukocyte counts were associated with higher hs-CRP levels (all *p* < 0.05), a larger CCAD, and a greater degree of concentric LV remodeling in terms of larger RWT in univariate models (all *p* < 0.05), except for lymphocyte counts. Only higher circulating monocytes were associated with larger LVMi (Table 2). After a multivariate adjustment, the total circulating WBC as well as lymphocyte and monocyte counts were all independently associated with a higher hs-CRP as well as a larger RWT and CCAD (all *p* < 0.05); however, only monocyte counts were associated with a higher LVMi (ß-Coef. 0.06, *p* = 0.01). The associations of WBC, segmented, and monocyte counts with a larger CCAD were more pronounced in men and obese subjects (all P _interaction_: < 0.05, Figure 2a), regardless of age. The relationship between various leukocyte count components and CCAD using RCS curves is illustrated in Figure 1. Overall, in this study, the total WBC, segmented, and monocyte counts, but not the lymphocyte count, showed positive relationships with CCAD.

### 3.3. Associations of Various Leukocyte Counts with CCAD and Clinical Outcomes

By using multivariate Cox linear regression and logistic regression models, we examined the prognostic influences of various leukocyte counts and CCAD at a median of 12.1 years (Interquartile range (IQR): 10.9–13.2 years) of follow-up. A total of 32 all-cause mortality and 158 HF events were observed, resulting in 175 event subjects (rate of events: 8.3%). A more advanced age, female sex, larger BMI, greater CCAD, higher leukocyte counts (including total WBC, segmented, and monocyte counts, but not lymphocyte count), and the presence of hypertension and CVD all independently predicted composite endpoint of all-cause mortality and HF admission using a backward stepwise regression analysis (all *p* < 0.05), with higher fasting sugar showing a borderline significance (Table 3). Figure 2b shows the risk of HF admission based on the CCAD and various leukocyte count fractions, with a greater CCAD, higher total WBC, segmented, and monocyte counts associated with higher HF risks.

We tested the prognostic values of various CCAD/leukocyte categories by categorizing various leukocyte counts into higher and lower groups (≤ 5.8 vs. > 5.8, ≤ 3.25 vs. > 3.25, ≤ 0.4 vs. > 0.4, and ≤ 1.88 vs. > 1.88 10^3^/µL for WBC, segmented, monocyte, and lymphocyte counts, respectively,). We observed significantly higher risks of composite HF/death in the higher CCAD subgroups in the combined CCAD/WBC, CCAD/segmented count, and CCAD/lymphocyte categories in univariate models (Table 4, Figure 3). After a full adjustment, subjects presenting both higher CCAD/WBC counts (adjusted HR (aHR): 1.98 (1.20, 3.28), *p* = 0.008) and higher CCAD/segmented counts (aHR: 2.18 (1.31, 3.62), *p* = 0.003) were independently associated with a higher risk of composite HF and all-cause mortality (Table 4). Significant and graded worse outcomes were observed with a higher CCAD and monocyte count, with a higher monocyte count aggravating the composite HF and all-cause mortality risk in subjects with greater CCADs (P_interaction_: 0.035). Those exhibiting both higher CCAD and monocyte counts showed nearly three-fold higher HF/death events after adjustment (aHR: 2.81 (1.57, 5.03), *p* < 0.001) (Table 3 and Table 4, and Figure 3).

## 4. Discussion

In this study, we demonstrated that greater carotid arterial remodeling was highly associated with aging, male sex, greater body size, the presence of hypertension, and higher leukocyte counts. Higher WBC, segmented, and monocyte counts rather than lymphocyte counts were associated with a higher hs-CRP, greater LV remodeling, and higher CCAD, which were more pronounced in men and in the obese subgroups. Only a higher monocyte count was related to a greater degree of ventricular remodeling and a greater indexed LV mass after multivariate adjustment. During follow-up, a greater CCAD, and higher WBC, segmented, and monocyte counts independently predicted HF development, especially in subjects manifesting both greater CCADs and higher monocyte counts.

Both hypertensive and senescence processes are widely known as traditional risk factors for HFpEF due to their persistent adverse effects on cardiovascular structural remodeling and functional alterations [17]. For example, senescence-related structural alteration in elastic arteries (e.g., common carotid artery) is typically characterized by a modest expansion of the vessel diameter as gradual “outward” yet hypertrophic (thickened) vascular wall remodeling as the specific phenotype, partly due to persistent elastin fiber decline [18]. Vascular smooth muscle cell (VSMC) proliferation and extracellular cell matrix deposition in the intima and media layers in response to chronically elevated arterial pressure and mechanical stress within the vessel wall from cyclic stretch over some years may lead to the impaired integrity of elastin fibers [18,19,20,21,22], which is accompanied by a proteoglycan/collagen matrix deposition, which causes greater vascular stiffness [23]. Notably, such a vascular remodeling and increased hemodynamic load assessed by CCAD has been shown to occur alongside pathological cardiac structural remodeling and subclinical myocardial dysfunction despite the preserved global pump function as expressed by HFpEF [6].

Apart from the effects of senescence and hypertension, several pathological inflammatory processes associated with metabolic derangements (i.e., central obesity or diabetes) also contribute to increased vascular stiffness. [9,24,25]. Low-grade inflammation in terms of elevated CRP and higher WBC or monocyte counts impairs vascular function, causing higher vascular stiffness, which is associated with a higher risk of hypertension [9,13,26]. These findings indicate a coupled pathophysiological basis in terms of systemic inflammation (i.e., increased WBC count and CRP), impaired endothelial function (i.e., reduced NO production via vascular smooth muscle stimulation), and subsequent vasculopathy [10,27]. Importantly, these adverse associations may even start in the pre-hypertensive stage, indicating a key pathological link between vascular stiffness and a state of inflammation [26,28]. These pathological processes are likely driven by activated macrophage infiltration, an enhanced level of metalloproteinases, and aggravated extracellular matrix degradation or elastolysis, resulting in the subsequent degradation of collagen and structural vascular enlargement [10,29]. Nevertheless, this pathological association has not been tested by utilizing CCAD measure [6].

Recently, the role of circulating monocytes, which are major inflammatory cells, has been identified in the pathogenesis [13,30] of heart failure through its specific signaling, which likely potentiates the cardiac inflammatory response (i.e., Toll-like receptors, or TLR4 expression) [31]. Recent data have shown that higher circulating monocyte counts likely mediate key pathological systemic inflammatory effects and play a role in a variety of chronic systemic diseases [32], including obesity and vasculopathy. The up-regulated monocyte chemoattractant protein 1 (MCP-1), the tumor necrosis factor alpha, and the transforming growth factor-β (TGF-β) [33] from activated circulating monocytes have been shown to contribute to myocardial fibrotic replacement [34]. On the other hand, it has been proposed that the innate inflammatory response is involved in expansive arterial remodeling through an increased matrix metalloproteinase (MMP) activity [35] (gelatinase MMP-2 or MMP-9) and the degradation of elastic fibers [36]. As a major precursor of macrophages that serve as sources of metalloproteinases at the tissue level, higher circulating monocytes may therefore theoretically aggravate these pathological processes [37]. Indeed, a higher monocyte count was independently associated with unfavorable ventricular remodeling and a greater CCAD in thr current study, suggesting the pathophysiological role of the circulating monocyte count in the ventricular-arterial remodeling process. Interestingly, the differential associations among various circulating white blood cell counts, hs-CRP levels, and vascular arterial (CCAD) or cardiac remodeling probably signify the diverse pathophysiological roles of the different WBC types mediating distinct biological effects at different levels. For example, a greater association of hs-CRP with the total WBC/segmented counts but a weaker association with the monocyte/lymphocyte counts was observed. We also observed higher associations of LVMi with the monocyte count than with other leukocytes, which probably suggests a more prominent biological role of monocytes at the tissue level.

To the best of our knowledge, the present study is the first to demonstrate the circulating monocyte count reflecting signal-specific elicited systemic inflammatory response as an independent determinant of greater degree CCAD remodeling, which likely serve as a clinical predictor of HFpEF in a large asymptomatic population. Recently, microvascular dysfunction in relation to heightened pro-inflammatory signaling from myocardial infiltrating leukocytes or macrophages and interstitial fibroblast proliferation have been identified as the main pathological mechanisms underlying HFpEF [10,11,38]. In this regard, the enlargement of the carotid artery can be viewed as an indicator of unbalanced carotid arterial tensile stress of the ventricular-arterial system. This may be caused by synergistic aging, a chronic hemodynamic load, and proteolytic/elastolytic activity from an up-regulated pro-inflammatory status, all of which are also potential markers of heightened cardiovascular risk [39,40,41] as part of “dilating arterial disease” syndrome [42,43].

Our current study had some limitations. It was performed at a time when a more comprehensive myocardial functional assessment (i.e., tissue Doppler imaging) was not feasible, and hence the associations were made between the comprehensive ventricular diastolic indices and CCAD with and without correcting the baseline characteristics. However, in our previous work using myocardial deformation imaging, we demonstrated that the myocardial function declines with increasing CCAD in a smaller sample size [6]. Furthermore, the current study failed to incorporate more novel circulating biomarkers targeting specific signaling with respect to metalloproteinases or extracellular matrix degradation. Thirdly, the relationship between CCAD and systolic HF (HFrEF) was not feasible in the current analysis due to the study setting. Further prospective studies in the future are necessary to address these limitations.

## 5. Conclusions

In a large asymptomatic middle-aged population with a preserved global cardiac pump function, higher leukocyte counts were strongly associated with the inflammatory marker, hs-CRP, and manifested distinct associations with unfavorable ventricular-arterial remodeling, mostly in men and obese populations. The independent associations among higher white blood cell counts, a larger carotid artery diameter, and a higher risk of heart failure remained after multiple adjustments, with higher monocyte counts manifesting a link with the accentuated risk of heart failure in subjects with a larger carotid artery diameter. Our findings demonstrated the specific biological roles of circulating monocytes in the pathogenesis of HFpEF.

## Figures and Tables

**Figure 1 diagnostics-10-00287-f001:**
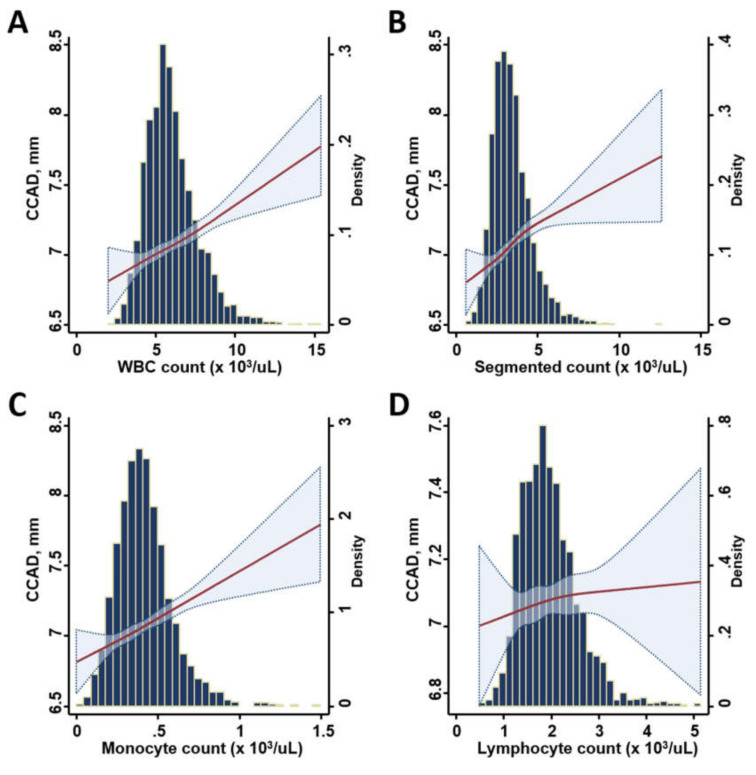
Restricted cubic splines (RCS) curves demonstrating the continuous relationship between white blood count fractions (including total WBC (**A**), segmented (**B**), monocyte (**C**), and lymphocyte counts (**D**) and common carotid artery diameter (CCAD). The y-axis displays the distribution and mean values of CCAD (mm).

**Figure 2 diagnostics-10-00287-f002:**
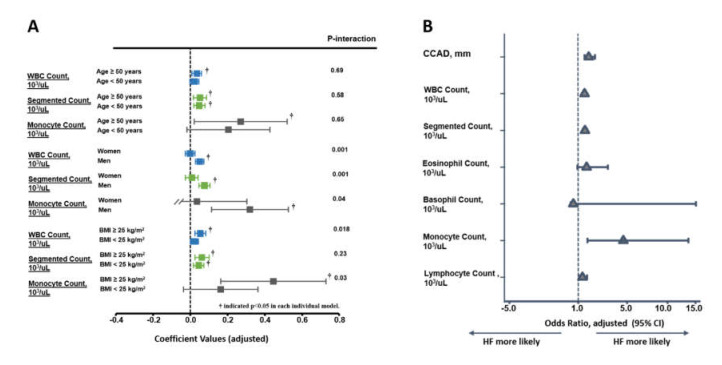
The associations between various leukocyte counts (including total WBC, segmented, monocyte, and lymphocyte counts) and common carotid artery diameter (CCAD) in the subgroup analysis (based on age (≥, < 50 years), sex, and BMI (≥, <25 kg/m^2^) categories) (**A**). The risks of HF admission based on CCAD and various leukocyte fractions after adjustment are presented as odds ratios and 95% confidence intervals (CIs) (**B**).

**Figure 3 diagnostics-10-00287-f003:**
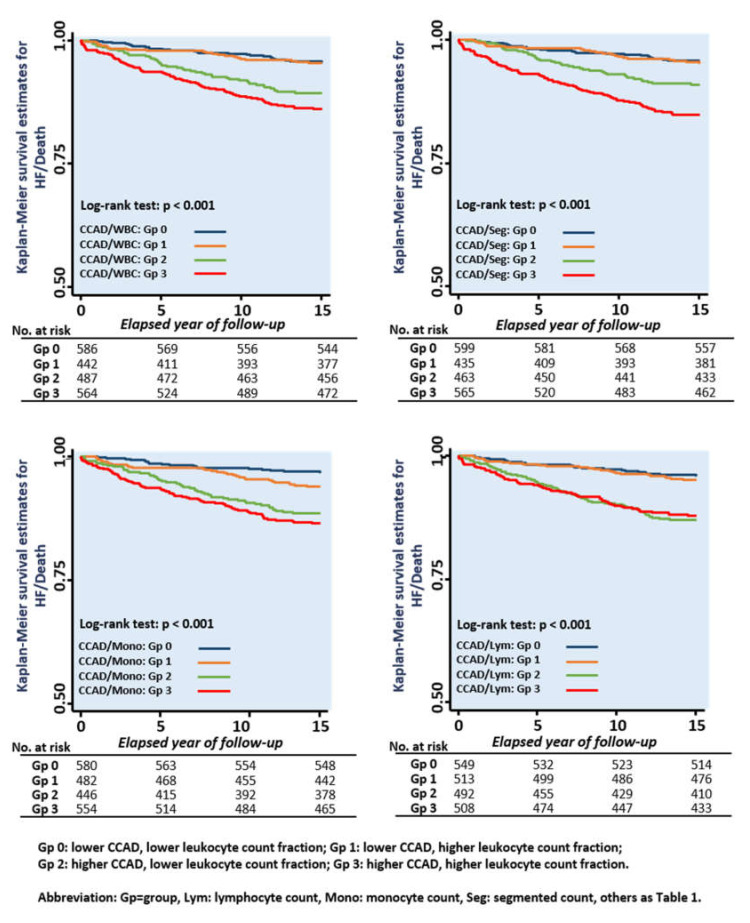
Kaplan–Meier curves demonstrating the associations of CCAD and various leukocyte count fraction categories (as lower and higher based on median values) with the composite HF and all-cause mortality risk.

**Table 1 diagnostics-10-00287-t001:** Baseline demographics and cardiac structural information according to common carotid artery diameter (CCAD) quartiles.

Metabolic Score Categories	All Subjects			CCAD Quartiles				*p* (Trend)
	(*n* = 2085)	Pearson (R)	*p* Value	Q1 (*n* = 546)	Q2 (*n* = 530)	Q3 (*n* = 506)	Q4 (*n* = 503)	
Baseline Demographics								
Age, yrs.	51.03 (10.73)	0.40	< 0.001	46.28 (9.15)	49.32 (9.34)	51.76 (9.94)	57.25 (11.34)	< 0.001
Female sex, *n* (%)	873 (41.20)	—	—	347 (63.55)	234 (44.15)	147 (29.05)	132 (26.24)	< 0.001
Systolic blood pressure, mm Hg	121.55 (17.55)	0.42	< 0.001	112.94 (14.62)	118.94 (15.46)	123.81 (15.90)	131.38 (18.75)	< 0.001
Diastolic blood pressure, mm Hg	75.51 (10.49)	0.31	< 0.001	71.05 (10.09)	74.70 (9.73)	77.29 (9.56)	79.43 (10.67)	< 0.001
Pulse pressure, mm Hg	46.05 (12.03)	0.34	< 0.001	41.89 (9.221)	44.24 (10.29)	46.52 (11.42)	51.92 (14.36)	< 0.001
Heart rate, min^−1^	74.71 (10.11)	0.02	0.47	74.39 (9.63)	74.41 (9.91)	75.19 (19.41)	74.90 (10.53)	0.246
Waist circumference, cm	82.37 (10.60)	0.39	< 0.001	76.86 (9.79)	80.69 (9.05)	84.85 (9.55)	87.62 (10.69)	< 0.001
Weight, kg	65.25 (12.27)	0.32	< 0.001	59.59 (10.44)	63.62 (10.61)	68.46 (12.43)	69.86 (12.78)	< 0.001
BMI, kg/m^2^	24.30 (3.65)	0.31	< 0.001	22.78 (3.15)	23.85 (3.19)	24.97 (3.58)	25.74 (3.94)	< 0.001
Body fat, %	26.85 (7.40)	0.04	< 0.001	26.67 (6.93)	26.88 (7.74)	26.58 (7.29)	27.30 (7.61)	0.277
Laboratory Data								
Fasting glucose, mg/dL	100.36 (23.77)	0.21	< 0.001	94.42 (15.69)	97.92 (20.46)	101.67 (22.83)	108.18 (31.81)	< 0.001
Total cholesterol, mg/dL	199.05 (37.68)	0.07	0.002	195.16 (35.67)	199.56 (40.96)	199.18 (32.81)	202.58 (40.42)	0.003
Triglyceride, mg/dL	136.15 (115.02)	0.15	< 0.001	113.50 (84.06)	132.20 (149.04)	141.96 (85.31)	159.14 (124.29)	< 0.001
HDL, mg/dL	55.30 (15.86)	−0.21	< 0.001	60.47 (17.05)	56.26 (15.48)	52.83 (14.21)	51.19 (14.84)	< 0.001
LDL, mg/dL	129.95 (33.15)	0.10	< 0.001	124.28 (32.15)	129.78 (32.40)	131.95 (29.84)	134.25 (37.13)	< 0.001
Uric acid, mg/dL	5.88 (1.48)	0.25	< 0.001	5.37 (1.38)	5.81 (1.38)	6.08 (1.43)	6.32 (1.55)	< 0.001
e-GFR, ml/min/1.73 m^2^	87.57 (17.69)	−0.17	< 0.001	91.13 (16.72)	88.08 (16.50)	87.84 (17.17)	82.86 (19.41)	< 0.001
Leukocyte Counts								
WBC count, 10^3^/µL	6.01 (1.62)	0.15	< 0.001	5.78 (1.48)	5.83 (1.58)	6.08 (1.61)	6.36 (1.77)	< 0.001
Segmented count, 10^3^/µL	3.43 (1.21)	0.15	< 0.001	3.27 (1.14)	3.26 (1.12)	3.52 (1.27)	3.69 (1.29)	< 0.001
Monocyte count, 10^3^/µL	0.42 (0.17)	0.15	< 0.001	0.39 (0.15)	0.41 (0.17)	0.43 (0.17)	0.45 (0.18)	< 0.001
Lymphocyte count, 10^3^/µL	1.96 (0.60)	0.03	0.22	1.94 (0.58)	1.95 (0.62)	1.94 (0.58)	1.99 (0.61)	0.15
Biomarkers								
hs-CRP (median, 25th–75th), mg/L	0.090 (0.043–0.210)	0.11	< 0.001	0.069 (0.030–0.155)	0.079 (0.040–0.165)	0.103 (0.050–0.230)	0.130 (0.070–0.270)	< 0.001
Nt-ProBNP (median, 25th–75th), pg/mL	28.05 (14.98–55.93)	0.15	< 0.001	31.15 (18.68–54.83)	26.95 (14.55–57.73)	22.60 (10.85–41.60)	33.55 (15.08–73.80)	< 0.001
Medical Histories								
Hypertension, *n* (%)	311 (14.68)	—	—	30 (5.49)	66 (12.45)	80 (15.81)	135 (26.84)	< 0.001
Diabetes, *n* (%)	113 (5.33)	—	—	14 (2.56)	23 (4.34)	27 (5.34)	49 (9.74)	< 0.001
CVD, *n* (%)	93 (5.63)	—	—	15 (3.52)	19 (4.49)	17 (4.12)	42 (10.77)	< 0.001
Regular exercise, *n* (%)	219 (21.88)	—	—	49 (18.49)	49 (19.68)	57 (24.57)	58 (26.01)	0.021
Active smoking, *n* (%)	187 (8.82)	—	—	37 (15.68)	44 (20.00)	40 (20.83)	62 (33.88)	< 0.001
Cardiac Structure and Function (*n* = 1805)								
IVS, mm	9.96 (1.53)	0.34	< 0.001	9.29 (1.39)	9.81 (1.48)	10.13 (1.41)	10.65 (1.52)	< 0.001
LVPW, mm	9.80 (1.39)	0.34	< 0.001	9.16 (1.26)	9.72 (1.34)	9.92 (1.26)	10.44 (1.38)	< 0.001
LVIDd, mm	46.67 (3.85)	0.30	< 0.001	45.18 (3.59)	46.22 (3.56)	47.18 (3.83)	48.19 (3.77)	< 0.001
LVEF, %	67.18 (4.84)	−0.05	< 0.001	67.41 (4.58)	67.31 (4.94)	67.29 (4.78)	66.70 (5.02)	0.038
RWT	42.52 (6.10)	0.18	< 0.001	40.99 (5.35)	42.49 (6.51)	42.73 (5.90)	43.96 (6.23)	< 0.001
LV mass, gm	152.62 (39.79)	0.43	< 0.001	131.27 (33.97)	146.99 (33.91)	157.94 (36.08)	175.83 (41.42)	< 0.001
LVMi, gm/m^2^	59.00 (15.80)	0.38	< 0.001	51.89 (13.45)	57.33 (14.28)	59.66 (14.11)	67.64 (17.12)	< 0.001
LVH, *n* (%)	278 (15.5)	—	—	38 (8.3)	61 (13.2)	68 (15.3)	111 (25.8)	< 0.001

Abbreviations: BMI, body mass index; CCAD, common carotid artery diameter; CVD, cardiovascular disease; e-GFR, estimated glomerular filtration rate; FS, fractional shortening; HDL, high-density lipoprotein cholesterol; hs-CRP, high sensitivity C-reactive protein; IVS, inter-ventricular septal wall thickness; LDL, low-density lipoprotein cholesterol; LVEF, left ventricular ejection fraction; LVH, left ventricular hypertrophy; LVIDd, left ventricular end-diastolic diameter; LVMi, indexed left ventricular mass; LVPW, left ventricular posterior wall thickness; Nt-proBNP, N-terminal pro B-type natriuretic peptide; RWT, relative wall thickness; WBC, white blood cell.

**Table 2 diagnostics-10-00287-t002:** Univariate and multivariate associations of various leukocyte counts with the inflammatory marker hs-CRP, carotid arterials, and ventricular structural parameters.

Independent Variables	hs-CRP, mg/L		CCAD, mm		LVMi, gm/m^2^ †		RWT	
Univariate (unit: 10^3^/µL)	ß-Coef.	*p* value	ß-Coef.	*p* value	ß-Coef.	*p* value	ß-Coef.	*p* value
WBC Count	0.25	< 0.001	0.07	< 0.001	0.03	0.15	0.1	< 0.001
Segmented Count	0.27	< 0.001	0.09	< 0.001	0.03	0.29	0.09	< 0.001
Monocyte Count	0.16	< 0.001	0.15	< 0.001	0.11	< 0.001	0.12	< 0.001
Lymphocyte Count	0.08	0.001	0.03	0.222	−0.005	0.84	0.06	0.008
Multivariate Model 1	ß-Coef.	*p* value	ß-Coef.	*p* value	ß-Coef.	*p* value	ß-Coef.	*p* value
WBC Count	0.24	< 0.001	0.04	< 0.001	0.03	0.26	0.09	< 0.001
Segmented Count	0.26	< 0.001	0.06	< 0.001	0.01	0.6	0.06	0.006
Monocyte Count	0.15	< 0.001	0.06	0.002	0.08	< 0.001	0.09	< 0.001
Lymphocyte Count	0.08	0.002	0.004	0.87	0.01	0.64	0.06	0.008
Multivariate Model 2	ß-Coef.	*p* value	ß-Coef.	*p* value	ß-Coef.	*p* value	ß-Coef.	*p* value
WBC Count	0.24	< 0.001	0.03	0.001	−0.01	0.73	0.06	0.009
Segmented Count	0.25	< 0.001	0.05	< 0.001	−0.01	0.52	0.05	0.037
Monocyte Count	0.15	< 0.001	0.05	0.007	0.06	0.01	0.08	0.001
Lymphocyte Count	0.07	0.007	0.004	0.88	−0.01	0.66	0.04	0.064

Abbreviations as Table 1. Model 1: adjusting for age, sex, and BMI. Model 2: adjusting for age, sex, BMI, systolic blood pressure, cholesterol, HDL, and medical history of hypertension, diabetes, cardiovascular disease, and estimated glomerular filtration rate (eGFR). † BMI was not included in the multivariate models.

**Table 3 diagnostics-10-00287-t003:** Stepwise multivariate Cox regression models for the composite endpoints of heart failure (HF) and all-cause mortality.

	Multivariate Cox Regression Models							
	(WBC Count)		(Segmented)		(Monocyte)		(Lymphocyte)	
	HR (95% CI)	*p* Value	HR (95% CI)	*p* Value	HR (95% CI)	*p* Value	HR (95% CI)	*p* Value
CCAD	1.33 (1.08, 1.65)	0.008	1.34 (1.08, 1.66)	0.007	1.33 (1.08, 1.65)	0.009	1.35 (1.09, 1.68)	0.006
WBC Count, 10^3^/µL	1.11 (1.02, 1.22)	0.018	—	—	—	—	—	—
*P _interaction_ (CCAD)*	—	0.56	—	—	—	—	—	—
Segmented Count, 10^3^/µL	—	—	1.15 (1.03, 1.29)	0.016	—	—	—	—
*P _interaction_ (CCAD)*	—	—	—	0.90	—	—	—	—
Monocyte Count, 10^3^/µL	—	—	—	—	2.56 (1.08, 6.04)	0.032	—	—
*P _interaction_ (CCAD)*	—	—	—	—	—	0.035	—	—
Lymphocyte Count, 10^3^/µL	—	—	—	—	—	—	1.10 (0.80, 1.51)	0.55
*P _interaction_ (CCAD)*	—	—	—	—	—	—	—	0.87
Age, +10 year	1.66 (1.40, 1.97)	< 0.001	1.65 (1.38, 1.96)	< 0.001	1.64 (1.38, 1.95)	< 0.001	1.63 (1.37, 1.95)	< 0.001
Sex (men), *n* %	0.65 (0.46, 0.91)	0.012	0.65 (0.46, 0.91)	0.011	0.62 (0.44, 0.88)	0.006	0.67 (0.48, 0.93)	0.017
BMI, +1 kg/m2	1.05 (1.01, 1.09)	0.023	1.05 (1.01, 1.09)	0.021	1.05 (1.01, 1.09)	0.02	1.05 (1.01, 1.10)	0.02
Hypertension	1.50 (1.03, 2.18)	0.036	1.48 (1.02, 2.17)	0.041	1.53 (1.05, 2.22)	0.027	1.55 (1.06, 2.26)	0.023
CVD	1.71 (1.05, 2.81)	0.032	1.70 (1.04, 2.79)	0.034	1.70 (1.04, 2.78)	0.033	1.69 (1.03, 2.75)	0.037
Glucose, +10 mg/dL	1.05 (0.99, 1.11)	0.089	1.05 (1.00, 1.11)	0.078	1.05 (1.00, 1.12)	0.062	1.06 (1.00, 1.12)	0.042

Abbreviations as in Table 1.

**Table 4 diagnostics-10-00287-t004:** Multivariate Cox regression models for the composite endpoints of HF and all-cause mortality based on the CCAD/leukocyte categories.

Cox Regression Models	CCAD and Leukocyte Counts (10^3^/µL) Categories			
CCAD/WBC Categories	CCAD ≤ 7, WBC ≤ 5.8	CCAD ≤ 7, WBC > 5.8	CCAD >7, WBC ≤ 5.8	CCAD > 7, WBC > 5.8
Crude HR	(Reference)	1.02 (0.58, 1.79), *p* = 0.96	2.45 (1.51, 3.96), *p* < 0.001	3.30 (2.11, 5.14), *p* < 0.001
Adjusted HR	(Reference)	0.94 (0.52, 1.69), *p* = 84	1.31 (0.77, 2.21), *p* = 0.32	1.98 (1.20, 3.28), *p* = 0.008
CCAD/Segmented Count Categories	CCAD ≤ 7, Seg ≤ 3.25	CCAD ≤ 7, Seg > 3.25	CCAD > 7, Seg ≤ 3.25	CCAD > 7, Seg > 3.25
Crude HR	(Reference)	1.13 (0.64, 2.01), *p* = 0.67	2.21 (1.34, 3.66), *p* = 0.002	3.85 (2.47, 6.02), *p* < 0.001
Adjusted HR	(Reference)	0.96 (0.53, 1.74), *p* = 0.90	1.22 (0.71, 2.10), *p* = 0.48	2.18 (1.31, 3.62), *p* = 0.003
CCAD/Monocyte Count Categories	CCAD ≤ 7, Mono ≤ 0.4	CCAD ≤ 7, Mono > 0.4	CCAD > 7, Mono ≤ 0.4	CCAD > 7, Mono > 0.4
Crude HR	(Reference)	1.95 (1.08, 3.51), *p* = 0.026	3.80 (2.22, 6.51), *p* < 0.001	4.56 (2.73, 7.64), *p* < 0.001
Adjusted HR	(Reference)	2.01 (1.09, 3.69), *p* = 0.025	2.38 (1.33, 4.24), *p* = 0.003	2.81 (1.57, 5.03), *p* < 0.001
CCAD/Lymphocyte Count Categories	CCAD ≤ 7, Lym ≤ 1.88	CCAD ≤ 7, Lym > 1.88	CCAD > 7, Lym ≤ 1.88	CCAD > 7, Lym > 1.88
Crude HR	(Reference)	1.22 (0.69, 2.16), *p* = 0.50	3.38 (2.08, 5.49), *p* < 0.001	3.15 (1.93, 5.13), *p* < 0.001
**Adjusted HR**	(Reference)	1.23 (0.68, 2.21), *p* = 0.49	1.87 (1.10, 3.17), *p* = 0.021	2.00 (1.18, 3.40), *p* = 0.01

Abbreviations as in Table 1 and as follows: Lym, lymphocyte; Mono, monocyte; Seg, segmented. Multivariate models were adjusted for age, sex, BMI, heart rate, medical histories of hypertension, diabetes and cardiovascular diseases, medication for hyperlipidemia, active smoking, regular exercise, and eGFR.

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
