# Peer review of "Circulating Monocyte Count as a Surrogate Marker for Ventricular-Arterial Remodeling and Incident Heart Failure with Preserved Ejection Fraction"

_diagnostics, 2020, doi:10.3390/diagnostics10050287_

Round 1
Reviewer 1 Report
Manuscript ID: diagnostics-786525
The manuscript titled “Circulating Monocyte Count as Surrogate Marker for Ventricular-arterial Remodeling and Incident Heart Failure with Preserved Ejection Fraction” by Wang K., et al., was aimed to study the possible association between the monocyte count with ventricular arterial remodeling and explore its putative prognostic value for HFpEF.
This is a well written manuscript. The authors utilized statistical analyses to establish the association between monocyte count and CCAD, and subsequently demonstrated its potential application as prognostic marker for HFpEF. The Reviewer would like to recommend the manuscript for publication with minor revision.
The authors may consider to further improve their manuscript by addressing following concerns,
- The authors may wish to proofreading their manuscript. For instance, there should be spaces between numbers and symbols (Line 27: 0 ± 10.7; Line 31.....) Typo: Line 168 Nt-proBNP: N-termina pro B-type natriuretic peptide.
- Full name of hs-CRP should be provided and stated in the main text.
- The authors reported that in addition to monocyte count, the associations between multiple parameters and known risk factors (such as uric acid, eGFR; hypertension, diabetes) with CCAD were also observed. The authors may wish to address or discuss the primary cause of the observed WBC/monocyte count alternations in PTs accordingly.
- Does the association observed in this study appear in HFrEF setting?The authors may wish to explore the possible association between leukocyte count and CCAD in HFrEF.
- The author may wish to revise “Our findings demonstrated the specific biological roles of leukocytes, particularly monocytes, in the pathogenesis of HFpEF.” in the conclusion section, as current study did not demonstrate the biological role of leukocyte in the pathogenesis of HFpEF.
Author Response
Response to Reviewer 1 Comments
1. The authors may wish to proofreading their manuscript. For instance, there should be spaces between numbers and symbols (Line 27: 0 ± 10.7; Line 31.....) Typo: Line 168 Nt-proBNP: N-termina pro B-type natriuretic peptide.
Response: we thank the reviewer’s comments on these. This work has been sent for extensive typo and grammatical errors correction, together with English language and style edition. These minor errors haven been corrected as much as possible accordingly. We further noticed that segment neutrophil count should be better changed as “segmented” neutrophil count, and hs therefore made extensive revision on this wording thoroughly (including tables and figures) in this work accordingly.
2. Full name of hs-CRP should be provided and stated in the main text.
Response: Yes, this has been added in Line 142, Page 3.
3. The authors reported that in addition to monocyte count, the associations between multiple parameters and known risk factors (such as uric acid, eGFR; hypertension, diabetes) with CCAD were also observed. The authors may wish to address or discuss the primary cause of the observed WBC/monocyte count alternations in PTs accordingly.
Response: Yes we thank the reviewer’s comment on this point. Indeed, several clinical risk factors are associated with greater CCAD. The pathological links between higher WBC/monocyte counts and higher vascular arterial stiffness and remodeling in terms of vascular enlargement have been demonstrated in prior reports (Line 286-299, page 5). We had addressed the possible causes for such relationship utilizing CCAD as vascular marker based on similar pathophysiological view point, which had been revised in Line 286-299, Page 5) accordingly.
We hope the reviewer appreciate and agree these explanations elaborated in our Discussion section.
4. Does the association observed in this study appear in HFrEF setting? The authors may wish to explore the possible association between leukocyte count and CCAD in HFrEF.
Response: Yes, I agree the reviewer’s comment on this point. Owing to data source, our analysis mainly comprised asymptomatic individuals underwent annual health check up free from prior or current HF history with preserved LVEF (a population suitable for exploring HFpEF). Therefore, this hypothesis (association between leukocyte count and CCAD in HFrEF) is less likely to be tested under current study setting.
However, prognostic impacts of elevated leukocyte counts in HFrEF (SHF) has been published in SOLVD study (Am J Cardiol.
1999;84:252-7). The relatively unexplored relationship between CCAD and HFrEF (SHF) was not feasible due to current study setting. Therefore we added this part in our study limitation section (Line 341-342, Page 6).
5. The author may wish to revise “Our findings demonstrated the specific biological roles of leukocytes, particularly monocytes, in the pathogenesis of HFpEF.” in the conclusion section, as current study did not demonstrate the biological role of leukocyte in the pathogenesis of HFpEF.
Response: Yes, we agree the reviewer’s point on this. We had since then re-phrased this sentence as “We conclude that a higher monocyte count was associated with cardiac remodeling and carotid artery dilation.” in the Abstract section, Line 38-38, Page 1. Another sentence “Higher circulating WBC count, segmented cell count, and monocyte count and greater CCAD were all independently associated with a higher risk of HF/all-cause death during a median of 12.1 years of follow-up in fully adjusted models” in Line 33-35, Page 1 was also updated based on findings from Table 3.
Finally, in Line 351-352, page 6, we further rephrased the sentence in the Conclusion section as “Our findings demonstrated the specific biological roles of circulating monocytes in the pathogenesis of HFpEF.”.
Reviewer 2 Report
The authors have used sound statistical methods to test the associations of carotid artery diameter with circulating leukocytes and inflammatory mediators.
My main concern is that the introduction and discussion do not really discuss the significance of the CCAD measurement. This, presumably, is a marker of vascular/arterial remodeling. Does it correlate with outcomes? Is it associated, in general, with HFpEF phenotypes? More description of this would allow us to understand the overall significance of the association of monocyte number with this parameter.
Another issue, is whether the authors also looked at more sensitive markers of diastolic function or filling pressures such as: E/e', E/A, or strain imaging on echocardiography. This information would be useful to discuss.
Author Response
Response to Reviewer 2 Comments
1. The authors have used sound statistical methods to test the associations of carotid artery diameter with circulating leukocytes and inflammatory mediators.My main concern is that the introduction and discussion do not really discuss the significance of the CCAD measurement. This, presumably, is a marker of vascular/arterial remodeling. Does it correlate with outcomes? Is it associated, in general, with HFpEF phenotypes? More description of this would allow us to understand the overall significance of the association of monocyte number with this parameter.
Response: Yes we thank the reviewer’s comment on this. CCAD alone was associated with outcomes in adjusted model and had been reported (Figure 2B, and Line 243-245, Line 248-249, Page 4). Our previous publication has already shown that greater CCAD relating to worsening cardiac mechanics by deformation measure as alternative HFpEF marker (reference number 6). This work mainly focused on the implications of CCAD in a relatively healthy population free from HF. As elicited systemic pro-inflammatory process and circulating inflammatory cells i.e. WBC or monocytes, reference number 9,13,26) have shown to mediate higher vascular arterial stiffness and dilation by activated macrophage infiltration, enhanced level of metalloproteinases, and aggravated extracellular matrix degradation or elastolysis, these association has not been tested by utilizing CCAD before. To clarify these issues in a better manner, we further revised Introduction (Line 46-54, Line 64, Page 2) and Discussion sections (Line 282-285, 298-299, 312-315, Page 5 and 6) to fully incorporate CCAD measure more clearly as part of our hypothesis better accordingly.
2. Another issue, is whether the authors also looked at more sensitive markers of diastolic function or filling pressures such as: E/e', E/A, or strain imaging on echocardiography. This information would be useful to discuss.
Response: Yes, indeed, we completely agree with the reviewer’s comment on this point. We also recognized this point as major limitation in current study and therefore had clearly addressed the design section under subtitle of Study Subjects, Materials and Methods as the following accordingly.
The data used for analysis were between January 2003 and June 2009. Part of the data has been published before, with the current study mainly focused on a study period prior to the use of advanced echocardiography imaging (i.e. tissue Doppler imaging). However, in our previous work, we also demonstrated that myocardial function declines using myocardial deformation imaging with increasing CCAD in a smaller sample size (reference number 6).
Therefore, the data in current analysis mainly focus on cardiac structure and remodeling information. limitations These had also been well described in Line 334-337, page 6 as study limitations accordingly.